# Spike-Event X-ray Image Classification for 3D-NoC-Based Neuromorphic Pneumonia Detection

**Jiangkun Wang** *[ID], **Ogbodo Mark Ikechukwu** [ID], **Khanh N. Dang** [ID] and **Abderazek Ben Abdallah** *[ID]

Graduate School of Computer Science and Engineering, The University of Aizu,
Aizu-Wakamatsu 965-8580, Fukushima, Japan
* Correspondence: ku@ieee.org (J.W.); benab@u-aizu.ac.jp (A.B.A.)

**Abstract:** The success of deep learning in extending the frontiers of artificial intelligence has accelerated the application of AI-enabled systems in addressing various challenges in different fields. In healthcare, deep learning is deployed on edge computing platforms to address security and latency challenges, even though these platforms are often resource-constrained. Deep learning systems are based on conventional artificial neural networks, which are computationally complex, require high power, and have low energy efficiency, making them unsuitable for edge computing platforms. Since these systems are also used in critical applications such as bio-medicine, it is expedient that their reliability is considered when designing them. For biomedical applications, the spatio-temporal nature of information processing of spiking neural networks could be merged with a fault-tolerant 3-dimensional network on chip (3D-NoC) hardware to obtain an excellent multi-objective performance accuracy while maintaining low latency and low power consumption. In this work, we propose a reconfigurable 3D-NoC-based neuromorphic system for biomedical applications based on a fault-tolerant spike routing scheme. The performance evaluation results over X-ray images for pneumonia (i.e., COVID-19) detection show that the proposed system achieves 88.43% detection accuracy over the collected test data and could be accelerated to achieve 4.6% better inference latency than the ANN-based system while consuming 32% less power. Furthermore, the proposed system maintains high accuracy for up to 30% inter-neuron communication faults with increased latency.

**Keywords:** spiking neural network; neuromorphic; reconfigurable; fault-tolerant; pneumonia; edge

## 1. Introduction

Artificial intelligence (AI) has in recent years been increasingly used across several fields. In healthcare specifically, deep learning (DL) models are being employed for applications such as timely detection of anomalies in patient health monitoring [1], lung ultrasonography classification [2], and most recently in the ongoing efforts to combat the raging COVID-19 pandemic through detection and diagnosis [3,4]. A common approach to meet computing requirements for such applications is to deploy DL models on cloud computing platforms [5]. However, some of these applications require secure real-time analysis of generated medical data [6–8], which subjects them to security [9] and latency [10] issues and makes them unsuitable for deployment. Although they still provide a viable approach to meet the low-latency [11], privacy-preserving [10], and security [12] requirements [13], edge computing devices find it challenging to meet the computation, memory, and power requirements of deep neural networks (DNNs) [14]. Despite conventional ANNs' impressive performance in various applications, they are computationally complex and power-hungry [15], thus making them less suitable to be deployed on edge computing platforms. However, the spiking neural network (SNN), which has been demonstrated to be more energy efficient [16] with the right information encoding scheme, can be used to achieve fast, energy-efficient, and real-time information processing for edge applications.

Compared to the conventional ANN, the computing principle of the SNN is more analogous to the brain, where communication among neurons is carried out in an event-driven manner. The operations of a simple leaky-integrate-and-fire (LIF) spiking neuron model can be characterized by the accumulation of weighted input, leak, and firing of action potentials triggered when the value of the membrane potential crosses a certain threshold. Given a sparse input spike train, the power consumption of SNNs is significantly reduced compared to ANNs. As shown in Figure 1, spike events are encoded in binary, and the processing involves only accumulation. Conventional ANN data are real static values that require multiply-accumulate (MAC) operations. SNNs, when implemented on hardware, tend to exploit parallelism and speed to achieve rapid and energy-efficient processing [17] for DL applications. The field-programmable gate array (FPGA) [18] is a prevalent hardware platform that has gained attention for edge computing due to its reconfigurability, cost, and energy efficiency.

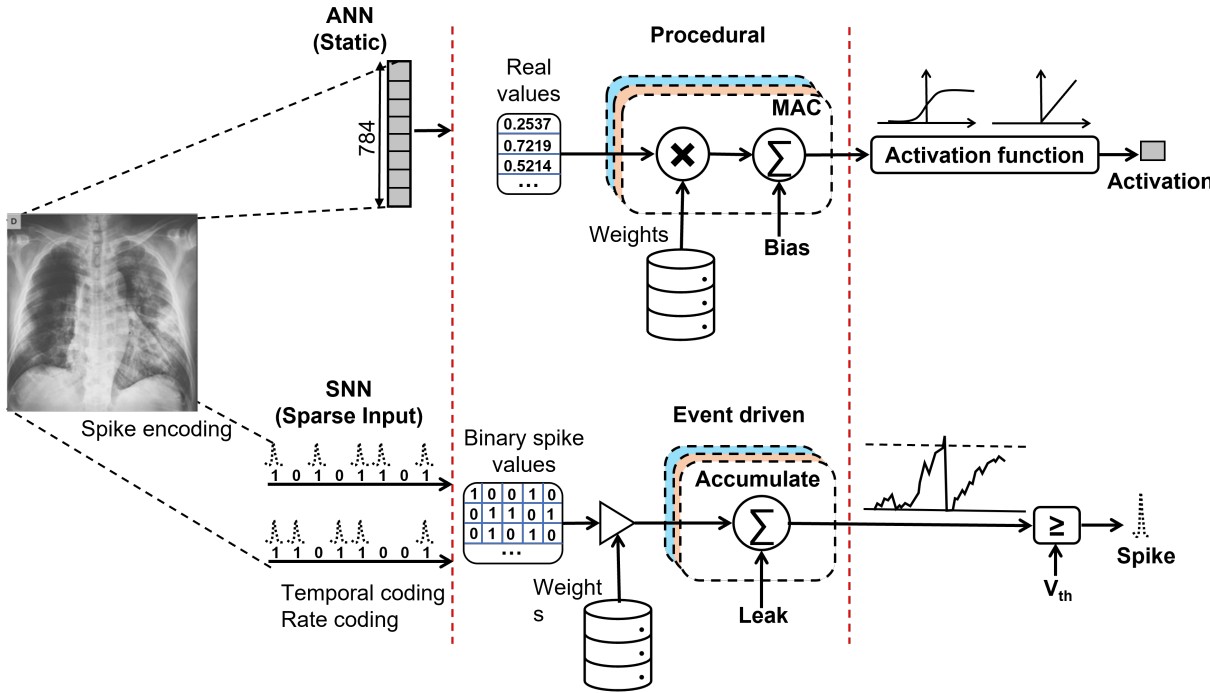

**Figure 1.** Comparison of computational between conventional ANNs and SNNs [19].

For the COVID-19 pandemic, which started at the end of 2019, countries worldwide have been channelling a lot of resources into many fields, including bio-medicine, with the hope of combating the menace strongly. With these efforts, the use of DL-based systems on edge computing platforms for rapid diagnosis/detection has drawn quite some interest in the biomedical engineering community. Deep learning models such as SNNs can be used to develop a reconfigurable 3D-NoC-based neuromorphic system suitable for pneumonia (COVID-19) detection for edge computing platforms.

*Background*

The computational complexity of deep learning models poses a challenge in edge applications; however, the brain's unique nature of information encoding makes it possible to achieve high energy efficiency. As the size of brain-mimic models such as SNN increases, the goal for effective communication among neurons becomes a challenge [20,21]. By integrating neuromorphic systems with 3D network-on-chip communication infrastructure, a highly scalable energy-efficient architecture suitable for biomedical applications can be realized.

By the end of 2019, the coronavirus disease caused by the COVID variant SARS-CoV-2, renamed COVID-19, began disrupting and affecting every aspect of life globally. As of August 2022, the number of cases reported is more than 578 million globally, with over 6 million deaths [22]. Detecting using efficient rapid diagnosis methods and isolating infected patients is one primary key to fighting the spread of the disease. The rapid diagnosis method uses the reverse transcription polymerase chain reaction (RT-PCR) technology which tests for genetic materials of the SARS-CoV-2 virus in upper respiratory specimens collected from patients. The sensitivity of this method ranges from 60% to 97% [23]. However, due to significant variations among different patients, the detection sensitivity is reduced to 60–71% [24]. Additionally, symptomatic patients and asymptomatic carriers may give false-negative results, which may mislead and pose a significant threat to public health safety.

Although test using this technology can be performed in batches, the samples to be tested is collected manually from test results. Another approach to test for the disease is to analyze lung X-ray images of patients, which has an accuracy range of 80 to 90% [25]. This approach requires doctors to examine the lung X-ray images of patients one at a time and combine the result with the patient's physical condition to complete a diagnosis. Nevertheless, with the rate of increase in the number of reported cases of COVID-19, this approach has become ineffective as it lacks quick response, reporting, and privacy security, which are essential requirements for patient treatment. In addressing these issues, computer-aided diagnosis systems that leverage deep neural networks (DNNs) have been considered as potential solutions [26,27]. However, typical medical institutions generally do not use power-efficient systems with enough computing resources for large-scale diagnoses. In addition, traditional biomedical information security measures conflict with distributed learning mechanisms making it challenging to aggregate diagnostic models of multiple medical institutions to improve detection accuracy and speed. Additionally, it becomes challenging to implement online reporting and response mechanisms to cooperate with the government. In this regard, clinicians and researchers have made great efforts to find alternative means to complement existing methods towards improving diagnostic accuracy in the detection/diagnosis of COVID-19.

DNNs consist of several densely connected layers, which enables them to achieve astounding variability that can be used for precise inference through training. Unfortunately, because they are also based on conventional ANNs, they are unsuitable to be used on edge computing platforms [28]. It is of essence to consider the need for efficient resource utilization [15], the goal towards achieving green AI [29], and low-power for satisfactory inference [30] while deploying DNNs for any application.

In our previous work, we proposed an AI-Enabled Real-time Biomedical System (AIRBiS-1) [31,32] for pneumonia (i.e., COVID-19) detection and health monitoring. It consists of a high-performance, reconfigurable inference AI chip with a robust collaborative-learning mechanism for privacy preservation and an interactive user interface for operation and effective monitoring. Although AIRBiS-1 achieved high accuracy in COVID-19 detection using a conventional ANN model, it is less suitable to be deployed for edge-based detection and monitoring due to the computational complexity and high power consumption of ANN models. In order to address this issue, we proposed in [33] an efficient pneumonia detection method in chest x-ray images based on a neuromorphic spiking neural network in software. This work extends our previous work by proposing a reconfigurable 3D-NoC-based neuromorphic system for biomedical applications based on [17] and evaluating it using X-ray images by performing COVID-19 detection/diagnosis.

The main contributions of this work are summarized as follows:

- A reconfigurable 3D-NoC-based neuromorphic system for biomedical applications based on a fault-tolerant spike routing mechanism.
- A comprehensive evaluation of the proposed system with a detailed comparison with other state-of-the-art approaches.

The rest of this paper is organized as follows. Section 2 reviews related works. Section 3 presents the proposed system. Section 4 provides comprehensive results from an evaluation of the proposed system, and in Section 5, we present the conclusion and future work.

## 2. Related Works

This section presents a survey of other approaches for computer-aided pneumonia detection. Figure 2 shows a summary of COVID-19 detection methods on various computing platforms.

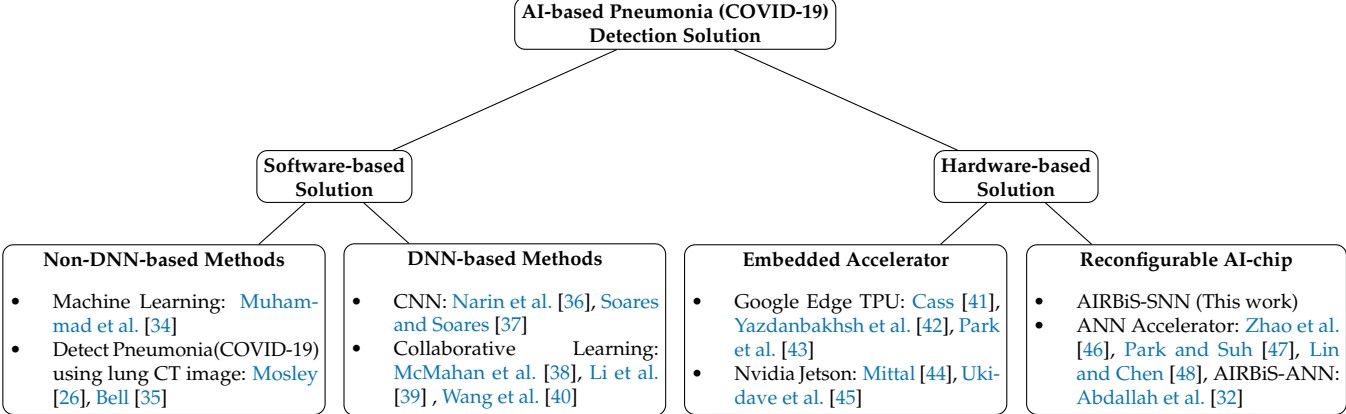

**Figure 2.** Summary of COVID-19 detection methods on various computing platforms [26,32,34–48].

### 2.1. Software-Based Approaches

Asraf et al. [27] surveyed the application of deep learning in the fight against COVID-19, including predicting protein structure, accelerating drug discovery, and infection detection based on medical images. The work in [49] collected a dataset, preprocessed it, and evaluated the accuracy of several CNNs (convolutional neural networks). Although the result of this study is promising, the method is limited to a small dataset, and the neural network model is not scalable. Mosley [26] described the RT-PCR and serological detection methods, which put huge pressure on the medical system and face cost and accuracy challenges. The origin, diagnosis, treatment, and other aspects of COVID-19 are discussed in [35]. The clinical manifestations of COVID-19 are extensive, but the symptoms and signs remain based on individual differences. A chest CT scan for infection detection is not recommended, but it may help detect complications. The work in [50] investigated the diagnostic value and consistency of chest CT scan and RT-PCR test methods in diagnosing COVID-19. Mustafa and Rahimi Azghadi [51] surveyed the development of AutoML technology and its applications in healthcare to assist the AI community in implementing automated learning of medical notes and reduce their over-reliance on human knowledge in the training process of machine learning.

### 2.2. Hardware-Based Approaches

To achieve rapid diagnosis/detection, other works have considered deploying accelerated DL diagnosis systems on edge platforms. A CT-based COVID-19 diagnosis and monitoring framework called ComputeCOVID19+ was proposed in [52] and accelerated across a multitude of heterogeneous platforms, including multi-core CPU, many-core GPU, and FPGA. The work in [53] proposed a COVID-19 detection algorithm using in-depth features and discrete social learning particle swarm optimization on edge platforms. The authors first used a pre-trained ResNet18 to extract features from CXR images and then used a discrete social learning particle swarm optimization algorithm to select features and a support vector to classify the images. A novel normalization algorithm using a CNN was proposed in [54] and implemented on FPGA to facilitate the pre-assessment of COVID-19.

## 3. System Architecture

A high-level view of the proposed (reconfigurable neuromorphic) biomedical system is described in Figure 3. It is based on our previously proposed fault-tolerant scalable 3D-NoC-based neuromorphic processor [55]. Its components include convolution cores, spiking neuron processing cores (SNPCs), and a fault-tolerant 3-dimensional router (FT-3DR) arranged in 2D mesh topologies and stacked to form a 3D mesh architecture. The convolution cores are tasked with extracting the features required for detecting pneumonia from X-ray images, while the SNPC analyzes the extracted features to detect if pneumonia is present or not. Communication between the convolution and spiking neuron processing cores is done via the FT-3DR. The rest of this section describes each of these components in detail.

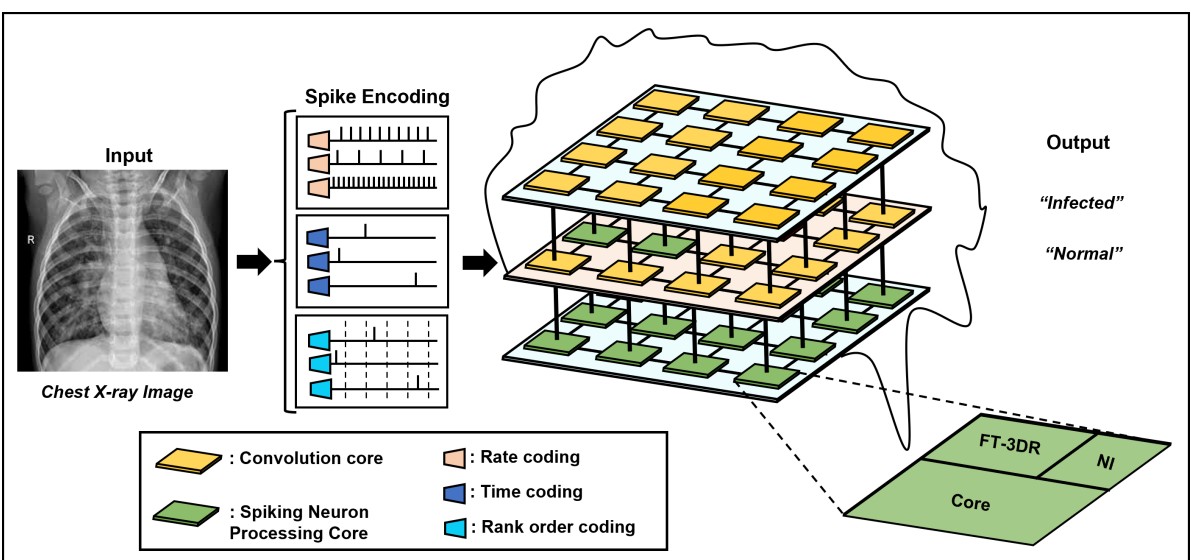

**Figure 3.** High-level view of the reconfigurable-reliable neuromorphic-based biomedical system.

### 3.1. Convolution Core

The architecture of the convolution cores is described in Figure 4. It consists mainly of a convolution and a pooling unit. The convolution contains a strider, kernel memory, and a controller. In contrast, the pooling unit contains a max pool unit, a register for storing the max, and a threshold for generating spikes from the accumulated max feature. A CXR image is first encoded into spikes using a single spike coding scheme and then sent to the convolution. At the convolution, the strider receives the spike-encoded CXR image over several time steps in 2-dimensional (2D) form and then flattens them. Using the size of the convolution filter, the strider selects from the flattened spike-encoded CXR image pixel values that match the position and size of a kernel stride. The selected pixel values are concatenated as an array such that their length is the same as that of a kernel when flattened. This length also corresponds to the depth of the kernel memories. One-hot operation is performed on the concatenated pixel values to determine pixels with spikes. The indexes of those pixels are then sent as memory addresses to the kernel memories to fetch the kernel weights stored at those addresses. After the weights are fetched, they are fed to the integrators. The number of these integrators corresponds to all kernels' total number of channels. When all the kernel weights associated with a single stride have been integrated, the integrated potential value is stored in a register representing the stride in the feature map. This process is repeated until all strides on the spike-encoded CXR image are completed, and a potential feature map for each kernel is generated. Afterward, the max pooling operation begins. All operations in the convolution core are managed by its control unit.

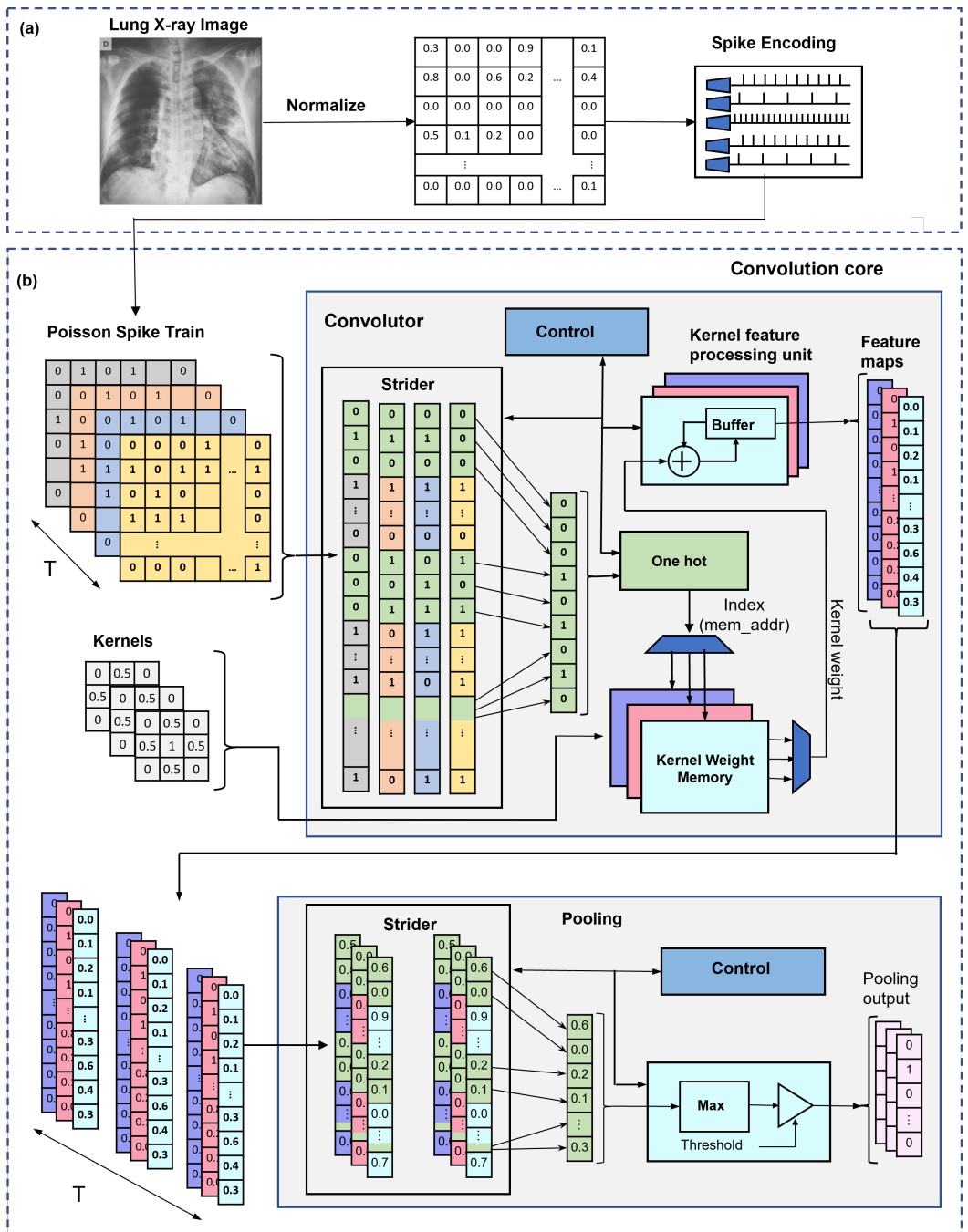

**Figure 4.** Convolution core architecture consisting of a convolutor and a pooling unit.

To perform the pooling operation, the strider performs a similar operation as with the convolution; however, the size of the pooling window is used to stride on the potential feature map. Within each pooling window, the potential with the highest value is added to the value of the potential register. The total value in the potential register is then compared with a threshold value, and if it exceeds a certain threshold, a spike is released and affixed to a position on the pooling operation output spike feature map corresponding to the stride. Afterward, the value of the potential register is reset to zero. No spike is generated if the potential value does not exceed the threshold. This operation is done in parallel on each potential feature map from the convolution. Depending on the SNN structure, if the next layer is a convolutional layer, the spike feature map generated by the pooling layer is sent to a convolution core, and the process is repeated until the next layer is fully connected to the spiking neuron processing core.

### 3.2. Spiking Neuron Processing Core

The SNPC is described in Figure 5, which is based on our previous work [17,55], consists of an array of leaky integrate and fire (LIF) neurons, a synapse crossbar, a synapse memory that represents the synaptic connection among the LIF neurons, and a control unit. The synapse memory stores the weights of the synaptic connection among the neurons, and the control unit manages the operations of the SNPC. When a spike is received at the SNPC, the spike feature map generated by the pooling layer is sent to the synapse crossbar, where it is analyzed for spike presence using a one-hot mechanism. If spikes are present, the indexes of the spikes in the spike feature map are used as the memory address of the associated synapses whose weights are stored in the synapse memory. Those synapse weights are then fetched from the synapse memory and sent to the corresponding post-synaptic neurons. The neurons accumulate the weighted input spikes and apply a leak operation that decays the value of the membrane potential. The accumulated resulting value is then stored in a register.

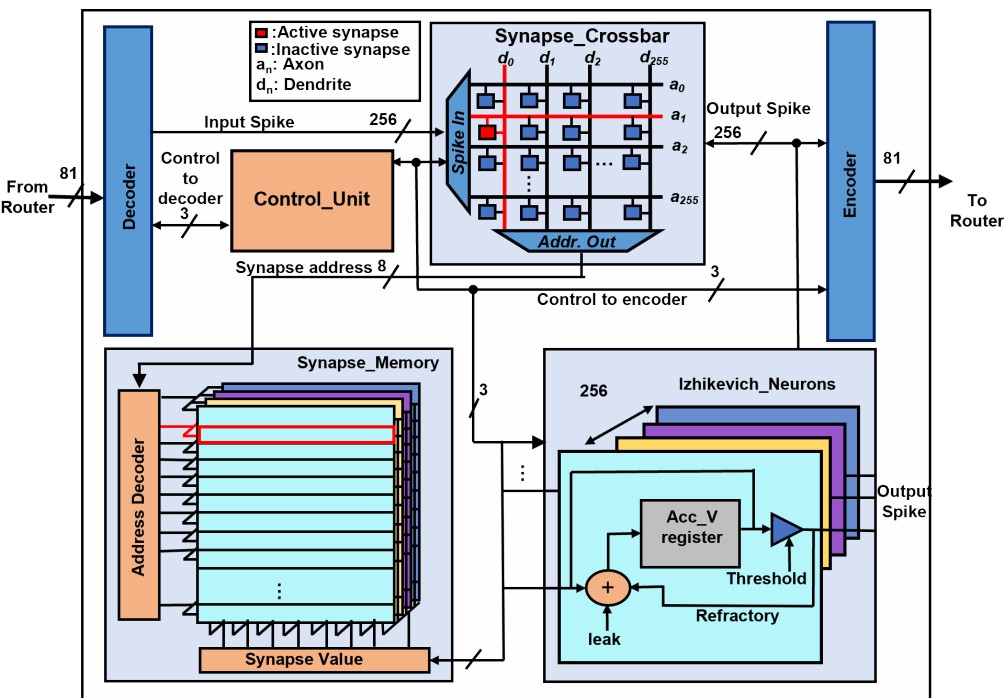

**Figure 5.** Architecture of spiking neuron processing core [56].

To mimic the leak current found in the neural membrane, the neuron receives a set leak value that causes decay in the membrane potential value when the leak is activated. When the value of the accumulated membrane voltage exceeds the set threshold, a 1-bit output spike is fired, and a signal is sent to the register to reset the value of the membrane voltage to zero and begin the refractory period. While in refractory, the neuron does not accumulate weighted input spikes. However, the refractory count gradually counts down every time step from the set refractory period to 0, and afterward, the neuron can accumulate weighted input spikes again. The accumulation of weighted presynaptic spikes as described in [57] is shown below:

$$V_j^l(t) = V_j^l(t-1) + \sum_i w_{ij} {}^* x_i^{l-1}(t-1) - \lambda \qquad (1)$$

where $V_j^l$ is the membrane potential of a LIF neuron $j$ in layer $l$ at a time-step $t$. $w_{ij}$ is the synaptic weight from neuron $i$ to $j$, $\lambda$ is the leak, and $x_i^{l-1}$ is a pre-synaptic spike from the previous layer $l-1$.

When input spikes are integrated, the value of $V_j^l$ at a timestep is compared with the threshold value $\theta$. If it exceeds, a spike ($s$) is released ($s = 1$), and the neuron resets. This is mathematically expressed in [17] as follows:

$$s = \begin{cases} 1, \text{if } V_j^l > \theta \\ 0, \text{if } V_j^l < \theta \end{cases} \tag{2}$$

### 3.3. Fault-Tolerant Multicast 3D Router

The fault-tolerant multicast 3D router, as described in Figure 6, routes spikes/packets in the system. Its design is based on [58,59]. The router has seven pairs of input–output ports. One pair connects to the local core, four pairs connect to neighboring routers in the north, east, south, and west direction using the intra-layer links, and the two remaining pairs connect to the up-and-down routers of the closest layers through TSVs. Each FTMC-3DR routes multicast packets in four pipeline stages. The first stage is buffer writing (BW), where the packet received is stored in the port's input buffer. The second pipeline stage is routing calculation (RC), where the source address of the stored packet is obtained and used to derive the address in the X, Y, or Z dimension. After this address is derived, the switch allocator (SA), which handles a stall/go flow control, and the matrix arbitration (matrix-arbiter scheduler) are triggered in the third stage to allocate the correct port to the next router or local SNPC. After the correct output port has been allocated, the fourth stage, crossbar traversal (CT), begins, where the packet traverses the crossbar to the allocated output port. Using a fault-tolerant shortest path k-means-based multicast routing algorithm (FTSP-KMCR) [59], the router can efficiently route packets from source to destination cores but is also able to handle permanent faults that may occur in the communication link between the cores. It does this by providing backup routes which enable faulty links to be bypassed during packet routing.

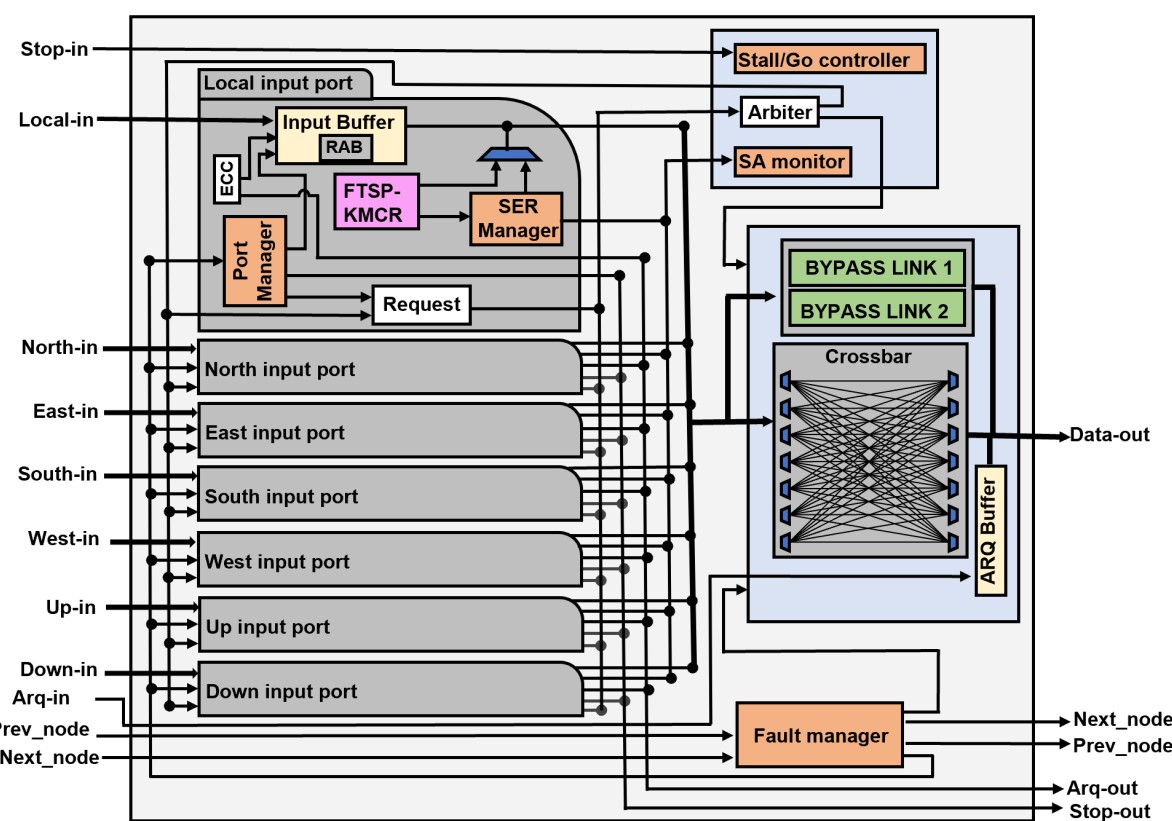

**Figure 6.** Architecture of fault-tolerant 3D router [55].

## 4. Evaluation

This section first provides the evaluation methodology for the proposed 3D-NoC-based neuromorphic pneumonia detection system. Then, it shows the detection accuracy on the chest X-ray dataset. In the following part, we evaluate the hardware complexity and fault-tolerant performance. Finally, we compare the results with existing works in terms of accuracy, inference time, and power consumption.

### 4.1. Evaluation Methodology

The proposed system is modeled and trained off-chip using back propagation spike-driven learning on the snnTorch simulation [60] framework with rate spike encoding. The rate encoding strategy is used to convert the X-ray images into time-varying spikes of different frequencies and to teach the final layer of the SNN how to respond to different input X-ray images. This way, the correct classes are encouraged to fire more frequently, and the incorrect classes to fire less. The gradient through the SNN is calculated backward through different paths from the loss to all synaptic weights and then summed before the weights are adapted. With the output spikes being discrete events, large perturbations of the membrane potential around the threshold are required to generate more spikes, which will keep the gradient of the SNN output at zero, and the synaptic weights cannot be adapted. However, by applying the target to the membrane potential, spiking can be promoted. This will ensure a strong gradient when there is a wrong classification. The hardware platform was designed on a Xilinx Zynq UltraScale+ MPSoC ZCU102 board using Verilog HDL, Xilinx's EDA suite (Vivado), and Cadence EDA tools. For ASIC implementation, the NANGATE 45 nm open-cell library [61] was used as the standard cells, OpenRAM [62] for generating the system memory, and TSV from FreePDK3D45 [63].

The SNN model summarized in Table 1 was trained using lung X-ray image dataset [64] from Kaggle [65]. The dataset is described in Table 2. It has 34,060 training images, of which 50.25% are COVID-positive and 42% are augmented (COVID (Augmented)) to increase the dataset size. After training, the trained weights were mapped on-chip for inference. To evaluate the proposed system for accuracy, average inference time, and fault tolerance, we use a lung X-ray image testing dataset with 1400 images, of which 50% are COVID-positive, as shown in Table 2. We also evaluate the design complexity regarding power consumption, area, and fault tolerance and compare the results with AIRBiS-1 [31] (ANN-based) and other existing works.

**Table 1.** Structure of SNN diagnosis/detection model for $64 \times 64$ input images.

| Layer | Output Shape | Parameters |
| --- | --- | --- |
| **Conv2D_1** | (16, 62, 62) | 160 |
| **MaxPooling2D_2** | (16, 31, 31) | 0 |
| **Leaky_3** | (16, 31, 31) | 0 |
| **Conv2D_4** | (32, 29, 29) | 4640 |
| **MaxPooling2D_5** | (32, 14, 14) | 0 |
| **Leaky_6** | (32, 14, 14) | 0 |
| **Flatten_7** | 12,546 | 0 |
| **Linear_8** | 2 | 12,546 |
| **Leaky_9** | (2)(2) | 0 |
| **Total parameters** | | 17,346 |
| **Trainable parameters** | | 17,346 |
| **Non-trainable parameters** | | 0 |

**Table 2.** Dataset description.

| Label | Class | Train | Test |
|---|---|---|---|
| **COVID** | COVID | 2870 | 700 |
| | COVID (Augmented) | 14,349 | - |
| **Non_COVID** | Normal | 9791 | 400 |
| | Lung_Opacity | 5762 | 250 |
| | Viral_Pneumonia | 1288 | 50 |
| Sum | | 34,060 | 1400 |

*4.2. Diagnosis/Detection Evaluation*

The SNN diagnosis/detection model, as summarized in Table 1, consists of two convolution layers, each having max-pooling and leaky layers along with a linear layer. The first convolution layer receives encoded lung x-ray images and extracts the required features according to the number of kernels. The extracted features are then sent to the max pool layer to reduce the size of the feature maps. Afterward, the output of the pooling unit is sent to the leaky layer, which encodes the max pool features into spikes by accumulating the values of the max pool features over various time steps and checking if they have exceeded a set threshold. The generated spike-encoded features are sent to the second convolution layer, which repeats the same process. The resulting spike feature from the second convolution layer is flattened and sent to the linear layer, which classifies them either as normal or infected.

To perform inference, the SNN model was mapped to a $3 \times 3 \times 3$ NoC architecture using a layer-layer-based mapping as described in Figure 7. With a total of 16 kernels, each having a single channel, the first convolution layer, together with its pooling and leaky operations, is mapped on the convolution cores in the first layer of the proposed system. Two kernels each were mapped to seven of the convolution cores and one kernel each on the two remaining cores. The second convolution layer, which consists of 32 kernels with 16 channels each, was mapped to the second layer of the system. Four kernels were mapped to seven convolution cores, making a total of twenty-eight kernels, and two kernels were mapped on each of the remaining two cores. Finally, the fully connected layer with two LIF neurons that classify the X-ray image as either infected or normal is mapped to two SNPC cores on the third layer of the proposed system. The SNN model achieved an inference accuracy of 88.43% with images encoded at 25 timesteps per X-ray image. The ANN-based system (AIRBiS-1) on the other hand, achieved an accuracy of 94.4%, as calculated using Equation (3) [31] below.

$$\text{Accuracy} = \frac{TP + TN}{TP + FN + FP + TN} \times 100\% \tag{3}$$

*4.3. Hardware Complexity Evaluation*

Table 3 shows the FPGA resource utilization of the proposed system alongside the ANN-based (AIRBiS-1) system. The proposed system utilized 9.9% of the lookup table (LUT), 1.28% of the LUTRAM, 6.7% flip flop (FF), and 4.45% of BUFG. Compared to the ANN-based system, the proposed system utilizes significantly fewer resources due to its reduced complexity in terms of binary input values, which enables lesser bandwidth and storage compared to the ANN-based system. Secondly, the proposed system requires no multiplication operation, which is in contrast to the ANN-based system.

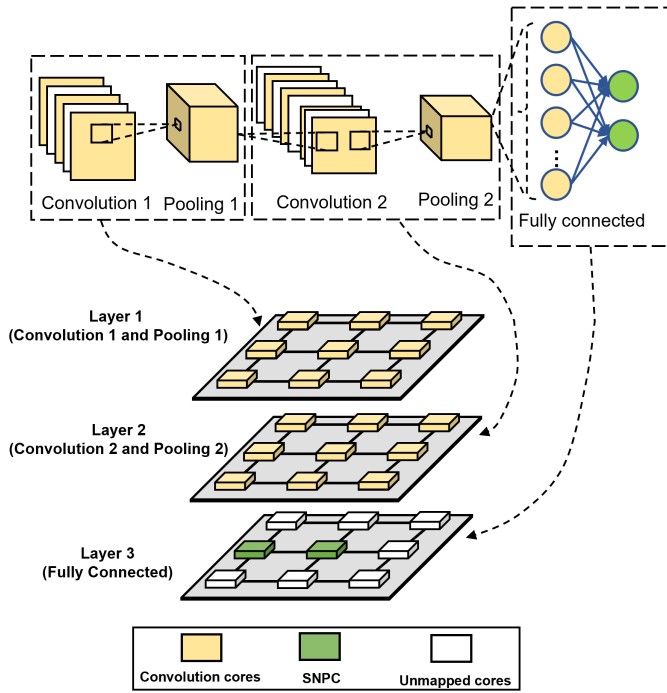

**Figure 7.** Layer-based mapping of the SNN model on the proposed system.

**Table 3.** FPGA resource utilization for the proposed neuromorphic system and the ANN-based system.

| Resource | Utilization | | Available | Utilization(%) | |
|---|---|---|---|---|---|
| | **ANN** | **SNN** | | **ANN** | **SNN** |
| **LUT** | 54,585 | 27,288 | 274,080 | 19.9 | 9.9 |
| **LUTRAM** | 3668 | 2048 | 144,000 | 2.5 | 1.28 |
| **FF** | 53,035 | 37,098 | 548,160 | 9.7 | 6.77 |
| **BRAM** | 824 | 0 | 912 | 90.4 | 0 |
| **DSP** | 35 | 0 | 2520 | 1.4 | 0 |
| **BUFG** | 4 | 18 | 404 | 1.0 | 4.45 |
| **MMCM** | 1 | 0 | 4 | 25 | 0 |

With an area of 0.0748 mm$^2$, the proposed system's convolution core has less area than the ANN-based system, which has 0.0755 mm$^2$. This difference in area is due to the inclusion of the multiply operation and the fixed point representation of input required for the ANN-based convolution. At the same time, the proposed system utilizes binary input and no multiplication operation. The power consumption of both convolution cores follows a similar trend, with the convolution core of the proposed system consuming less power at 0.007 mW compared to the ANN-based, which has 0.011 mW.

SNPC fully connects the proposed system's layers, while a multiply-accumulate (MAC) core connects the ANN model for the ANN-based system. For a fair comparison, we included 256 MAC units in the MAC core to match the 256 LIF neurons of the SNPC. The area and power consumption of the proposed system are 1.325 mm$^2$ and 0.007 mW, respectively; however, they are significantly less than that of the ANN-based system, which is 1.434 mm$^2$ and 0.011 mW, respectively. This trend also occurs due to the higher complexity in the ANN-based system's MAC unit compared to the proposed system's LIF unit.

To classify a single X-ray image at 100 MHz, the proposed system utilizes an average inference time of 41 ms. As shown in Figure 8, 35.9% of this time is utilized by the first

convolution layer, 7.9% by the second convolution layer, 6.6% by the SNPC, and 49.6% when receiving the input data. As seen in Figure 9a, inference with a smaller number of timesteps yields lower accuracy, and even increasing the number of timesteps beyond 25 does not increase the accuracy. For the ANN-based system, it takes about 43 ms to classify an X-ray image at 100 MHz, which is a higher classification latency compared to the proposed system.

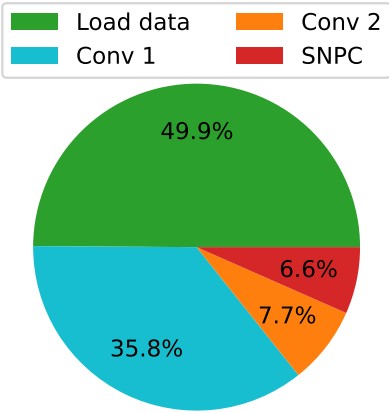

**Figure 8.** Average classification time complexity.

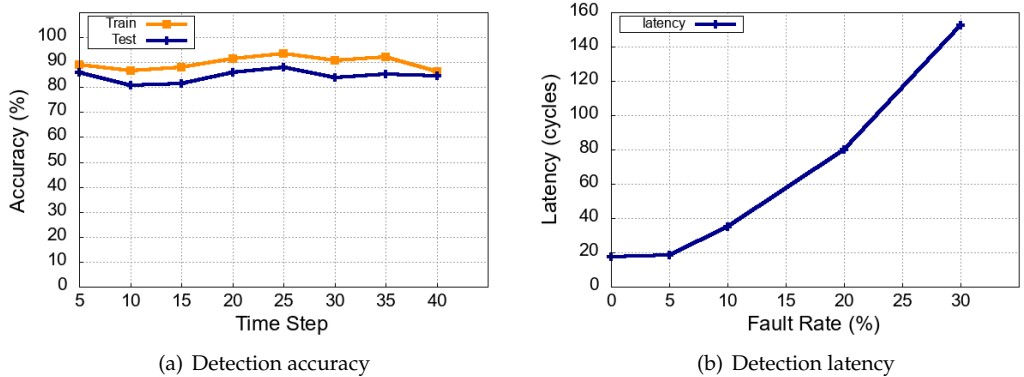

(a) Detection accuracy      (b) Detection latency

**Figure 9.** Pneumonia detection on the proposed system with various simulation time steps and fault rates. (**a**) Detection accuracy over various time steps. (**b**) Spike communication latency over various fault rates.

Figure 9b describes the fault-tolerant evaluation of the proposed system. The simulation was performed using a spike injection rate of 9. This is because using a lower spike injection rate requires spike packets from all nine layer cores to be sent at once over a shorter cycle, which will cause some packets to be dropped. The proposed system can maintain its detection accuracy with up to 30% of fault in the communication links. However, spike communication latency increased drastically as the fault rate increased from 5% to 30% due to increased spike packets that had to bypass the faulty link to take the backup path. However, the increase in fault rate does not affect the throughput of the system, which remains the same irrespective of the fault rate. Exceeding a fault rate of 30%, however, causes many packets to be dropped as there are not enough paths to get packets to their destination.

### 4.4. Comparison with Existing Works

Table 4 shows a detailed comparison of results obtained from the proposed system with other existing works. The work in [66] utilized an SNN with an X-ray image size of 256 × 256 and achieved an accuracy of 78%. In [33], an SNN was also utilized with

X-ray images of size 64 × 64 and achieved an accuracy of 80.7%. Both works, however, achieved lower accuracy compared to the proposed work, which is 88.43%. The difference in accuracy could be attributed to the SNN architecture, which is smaller for both works. Furthermore, both systems were implemented in software and did not provide information about power consumption. The work in [67] utilized 224 × 224 X-ray images. However, an ANN model was used, and it achieved less accuracy than the proposed system. Compared to our previously proposed work AIRBiS-1 [31], which is the same as the evaluated ANN-based model that we evaluated, a similar dataset was used, and a higher accuracy of 94.4% was achieved. However, the proposed system consumes less power, area, and lower inference time, as described in Section 4.3.

**Table 4.** Comparison with the existing studies of the neuromorphic system.

| Works | Model | Platform | Dataset | Image Size | Accuracy (%) |
|---|---|---|---|---|---|
| Fukuchi et al. [33] | SNN | Software | X-ray | 64 × 64 | 80.7 |
| Kamal et al. [66] | SNN | Software | X-ray | 256 × 256 | 78 |
| Che et al. [67] | ANN | Software | X-ray | 224 × 224 | 71.9 |
| AIRBiS-1 [31] | ANN | FPGA | X-ray | 256 × 256 | 94.4 |
| This work | SNN | FPGA | X-ray | 64 × 64 | 88.43 |

## 5. Conclusions

This work presents a reconfigurable fault-tolerant 3D-NoC-based neuromorphic system for biomedical applications targeted for pneumonia (i.e., COVID-19) detection in chest X-ray images. The system achieves energy-efficient information processing and fault tolerance by leveraging the event-driven information processing in SNNs and relying on a fault-tolerant spike routing scheme. Evaluation results show that the proposed system consumes 32% less power and requires about 4.6% less time for inference with a minor classification accuracy degradation than the ANN-based system. Compared with other existing works, the proposed system achieves higher accuracy and can remain functioning with up to 30% inter-neuron communication faults with increased latency. With the increasing need for continuous adaption to the dynamic user and environment features, progressive task exigences, and privacy, learning on the edge has become imperative. This has prompted the need for neuromorphic systems deployed for edge applications to support on-chip learning. The proposed system, however, currently does not enable on-chip learning, and addressing this will provide a guideline for future work, which would require exploring learning algorithms that would enable efficient on-chip learning. Considering that the main aspects of designing neuromorphic systems involve methods that effectively exploit the spatio-temporal and sparse activation feature of SNNs, the choice of the spike encoding scheme, which has been shown to affect the classification accuracy, robustness, processing latency, synaptic operations, area, and power consumption of a neuromorphic system, has to be rightly chosen [68]. Choosing the right spike encoding scheme, however, depends on the top requirements of an application, which could be accuracy, resilience to noise, network compression, or robustness. Therefore, future works will also include exploring other methods of spike encoding schemes that could improve the system's accuracy and overall system efficiency.

## 6. Patents

A. Ben Abdallah, H. Huang, N. K. Dang, J. Song, "AI Processor", Nbr. 2020-194733, Provisional Patent, 24 November 2020.

**Author Contributions:** Conceptualization, J.W., O.M.I. and A.B.A.; Data curation, J.W. and O.M.I.; Formal analysis, J.W.; Funding acquisition, A.B.A.; Investigation, J.W. and O.M.I.; Methodology, J.W., O.M.I. and K.N.D.; Project administration, J.W. and A.B.A.; Resources, J.W.; Software, J.W. and O.M.I.; Validation, J.W. and O.M.I.; Writing—original draft, J.W. and O.M.I.; Writing—review and editing, J.W., O.M.I, K.N.D. and A.B.A. All authors have read and agreed to the published version of the manuscript.

**Funding:** This work is partially supported by the University of Aizu Competitive Research funding, Ref. 2021-P6 and 2022-P7.

**Institutional Review Board Statement:** Not applicable.

**Informed Consent Statement:** Not applicable.

**Conflicts of Interest:** The authors declare no conflict of interest.

## Abbreviations

The following abbreviations are used in this manuscript:

| | |
|---|---|
| AIRBiS | AI-Enabled Real-time Bio-System |
| AI | Artificial Intelligence |
| ANN | Artificial Neural Network |
| AXI | Advanced eXtensible Interface |
| BRAM | Block Random-Access Memory |
| BW | Buffer Writing |
| CADx | Computer-Aided Diagnosis |
| CLB | Configurable Logic Block |
| CNN | Convolutional Neural Network |
| COVID-19 | Coronavirus Disease 2019 |
| CPU | Central Processing Unit |
| CXR | Chest X-ray |
| DL | Deep Learning |
| DMA | Direct Memory Access |
| DNN | Deep Neural Network |
| DSP | Digital Signal Processing |
| FF | Flip Flop |
| FLOPs | Floating-Point Operations |
| FPGA | Field-Programmable Gate Array |
| FT-3DR | Fault-Tolerant 3-Dimensional Router |
| FTSP | Fault-Tolerant Shortest Path |
| GOPS | Giga Operations Per Second |
| GPU | Graphic Processing Unit |
| GUI | Graphical User Interface |
| IID | Independent and Identically Distributed |
| IoT | Internet of Things |
| KMCR | K-means-based Multicast Routing Algorithm |
| LUT | Look-Up Tables |
| MACC | Multiply-Accumulate operations |
| NoC | Network-on-Chip |
| PL | Programmable Logic |
| PS | Processing System |
| RAM | Random Access Memory |
| ReLU | Rectified Linear Units |
| SNN | Spiking Neural Network |
| SNPC | Spiking Neuron Processing Core |

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
