# Peer review of "Spike-Event X-ray Image Classification for 3D-NoC-Based Neuromorphic Pneumonia Detection"

_electronics, doi:10.3390/electronics11244157_

Round 1

Reviewer 1 Report

The present paper proposes a 3D-NoC-based neuromorphic system for biomedical applications which is reconfigurable and is based on a fault-tolerant spike routing scheme. The authors stated higher accuracy for the proposed system.

The literature review does not cover approaches for edge computing well, especially IoT-based state-of-the-art approaches. The paper is well-written, however, there are following concerns.

11) Why is the comparison based on baseline (ANN-based)? Why not state-of-the-art approaches, because improving over baseline approach is not contributing since there is much research on this problem?

2) Why in table 3 the utilization of some resources for SNN missing? Why is BUFG more than four times worse than SNN?

3) Table 4 is based on which dataset? As I see, the datasets are different, which is not a fair comparison.

4) How do the experiences and performance show generalization on edge platforms with different limitations?

Reviewer 2 Report

Dear Authors, 

Please consider the following remarks:

1. In [16] authors suggest that the challenge of using SNNs for machine learning tasks, is in their training. Could you refer to that and explain in more details how the SNN was trained? Have you found any efficiency issues during that process?

2. In the equation (2) please write the full equation for clarity:

…. = 1, , if Vj l > θ or … = 0, otherwise 

3. Please add the reference in the text to equation (3).

4. Line 255: The SNN model summarized in Table 2 was trained using lung X-ray image dataset [65, 66] from Kaggle. The dataset is described in Table 1.

Comment: the Tables should be mentioned in the same order that they appear in the article. What does ‘COVID (aug)’ in Table 1 refer to?

5. Could you explain in more details the difference in accuracy between the work in [68] and this work? Can the speed of the systems be compared?

6. Can you comment on the AIRBiS-1 system proposed in [67] with comparison to your work. How both systems can be further developed to optimize the accuracy, power and inference time?

Best regards, 

Reviewer 3 Report

This work presents a reconfigurable fault-tolerant 3D-NoC-based neuromorphic system for biomedical applications targeted for pneumonia (i.e., COVID-19) detection in Chest X-ray images. The system achieves energy efficient information processing and fault tolerance by leveraging the event-driven information processing in SNNs and by relying on a  fault-tolerant spike routing scheme respectively. Evaluation results show that the proposed  system consumes 32% less power, requires about 4.6% less time for inference with a minor classification accuracy degradation than the baseline system. When compared with other  existing works, the proposed system achieves higher accuracy and can remain functioning with up to 30% inter-neuron communication faults with increased latency. So the scope of the paper considers the very important topic. The results are clear and good for research paper. Electronics journal is adequate journal for this type of research.

But some revision is required before final acceptance. 

-> please reorder the figure 2 and the text. In present form between ref 19 and 37 is a gap.  I know the missing refences are included at figure 2 but at the present form it's not correct.

-> for the equations given in paper eg (1) please add the adequate literature source. 

-> between 4. Evaluation and 4.1. Evaluation Methodology please add a general introduction to section 4.

-> please discuss the limitation of the proposed approach in section 5 - conclusions.

-> I familiarize with future works that are indicated " we intend to explore further other functionalities like learning algorithms and information coding schemes which could improve the accuracy of the system." and I believe that authors are able to discuss deeper the future dirrections.

Round 2

Reviewer 1 Report

The improvements made the paper clear and the comparisons are satisfactory after revision.